# Effectiveness of S-O Clip-Assisted Colorectal Endoscopic Submucosal Dissection

**DOI:** 10.3390/jcm11010141

**Published:** 2021-12-27

**Authors:** Haruka Fujinami, Akira Teramoto, Saeko Takahashi, Takayuki Ando, Shinya Kajiura, Ichiro Yasuda

**Affiliations:** 1Department of Endoscopy, Toyama University Hospital, Toyama 930-0194, Japan; 2Third Department of Internal Medicine, University of Toyama, Toyama 930-0194, Japan; akira_teramoto@hotmail.com (A.T.); aec56260@hotmail.co.jp (S.T.); taando33@gmail.com (T.A.); yasudaich@gmail.com (I.Y.); 3Department of Clinical Oncology, University of Toyama, Toyama 930-0194, Japan; shin-ya@nsknet.or.jp

**Keywords:** endoscopic submucosal dissection, colorectal tumor, traction method

## Abstract

This study aimed to assess the utility of the S-O clip during colorectal endoscopic submucosal dissection (ESD). We conducted a retrospective study on 185 patients who underwent colorectal ESD from January 2015 to January 2020. The patients were divided into two groups: before and after the introduction of the S-O clip. Forty-two patients underwent conventional ESD (CO group) and 29 patients underwent ESD using the S-O clip (SO group). We compared the surgery duration, dissection speed, en bloc resection rate, and complication rate between both groups. Compared with the CO group, the SO group had a significantly shorter surgery duration (70.7 ± 37.9 min vs. 51.2 ± 18.6 min; *p* = 0.017), a significantly higher dissection speed (15.1 ± 9.0 min vs. 26.3 ± 13.8 min; *p* < 0.001), a significantly higher en bloc resection rate (80.9% vs. 98.8%; *p* ≤ 0.001), and a significantly lower perforation rate (4.3% vs. 1.3%). In the right colon, the surgery duration was significantly shorter and the dissection speed was significantly higher in the SO group than in the CO group. Moreover, the rate of en bloc resection improved significantly in the right colon. S-O clip-assisted ESD reduces the procedure time and improves the treatment effects, especially in the right colon.

## 1. Introduction

Endoscopic submucosal dissection (ESD) is an established treatment for intramucosal tumors of the gastrointestinal tract, including the colon and rectum. This method, compared to conventional endoscopic mucosal resection (EMR), enables en bloc resection of larger lesions and has a low recurrence rate of 0.4–1.0% [1,2]. Colorectal ESD has several limitations, including an anatomically difficult procedure, a longer procedure compared to that of endoscopic mucosal resection, and a high risk of perforation and bleeding [2,3,4,5]. Moreover, some studies have reported life-threatening complications, such as perforation, the incidence of which was 4.1–5.3% [6,7].

Performing traction-assisted ESD will be easier if the submucosal layer can be directly visualized after the mucosal cut. Several traction techniques on lesions have been reported to be effective during ESD for large early gastric and colorectal cancers [8]. In particular, the S-O clip (TC1H05; Zeon Medical Co., Ltd., Tokyo, Japan) has been reported to be safe to use and to hasten colorectal ESD [9,10]. This study aimed to assess the utility of colorectal ESD using the S-O clip.

## 2. Materials and Methods

### 2.1. Patients

In this retrospective study, medical record of all patients who underwent colorectal ESD at the Toyama University Hospital were reviewed. Patients were divided into two groups according to the date of ESD procedure as S-O clip (Figure 1) were introduced in May 2017: the CO group underwent conventional ESD from September 2015 to April 2017, and the SO group underwent S-O clip-assisted ESD from May 2017 to January 2020. The indications for ESD included (1) a colonic neoplasm (adenoma and carcinoma) measuring >20 mm that was difficult to resect en bloc by conventional EMR, (2) a suspicion of an intramucosal lesion, and (3) the absence of submucosal invasion on magnifying endoscopy. Patients were excluded if they (1) had a rectal lesion, (2) showed a non-lifting sign or had residual lesions after endoscopic resection, or (3) had a lesion measuring >50 mm. All procedures were performed by five endoscopists who had performed more than 20 gastric ESD procedures.

### 2.2. ESD Preparation

A single-channel endoscope with a water-jet function (PCF-Q260AZI or PCF-H290ZI; Olympus Optical Co., Ltd., Tokyo, Japan) was prepared for the ESD. A transparent hood was attached to the endoscope to provide sufficient space and facilitate submucosal dissection. A solution containing sodium hyaluronate (MucoUp; Boston Scientific Co., Tokyo, Japan), saline, and a small quantity of indigo carmine were injected into the submucosal layer. As the border of the colonic neoplasm was generally clearly visible, no marking was carried out. Carbon dioxide (CO_2_) was used in all cases for insufflation.

Using the Jet-B knife (BSJB15B; Zeon Medical Co., Ltd.) and the SB Knife Jr (MD-47703; Sumitomo Bakelite, Tokyo, Japan), a circumferential mucosal incision was made and submucosal dissection was performed. Hemostasis was performed using Coagrasper (FD-411QR; Olympus Optical Co., Ltd.) with an electric surgical unit (VIO 300D; ERBE, Tübingen, Germany). The electrical power setting for the Jet-B knife was as follows: (1) dry-cut mode, effect 2, 50 W for the mucosal incision; and (2) forced-coagulation mode, effect 2, 50 W for the submucosal dissection. The setting for the SB Knife Jr was endo-cut Q-mode, effect 1, duration 1, interval 1, and that for the soft-coagulation mode was effect 4, 40 W for hemostasis. The setting for the Coagrasper was the soft-coagulation mode, effect 2, 40 W. All procedures were recorded on DVDs.

### 2.3. Conventional Colonic ESD

Using the Jet-B knife, an initial mucosal incision was made on the anal side of the lesion, followed by submucosal dissection. Next, the mucosal incision was extended to the right and to the left, and the submucosal layer under the extended area was dissected. When hemorrhage occurred during surgery, hemostasis was achieved using the Jet-B knife in the forced-coagulation mode or by using the Coagrasper [11]. In technically difficult situations, the SB Knife Jr was used thanks to its safety and usefulness [12,13]. Mucosal incisions and submucosal dissections were repeated; then, circumferential mucosal incisions and submucosal dissections were performed.

### 2.4. S-O Clip-Assisted ESD

First, a circumferential incision on the mucosal layer was performed using the Jet-B knife or the SB Knife Jr. Then, the S-O clip was attached to the proximal edge of the lesion (Figure 2A,B). Another clip was used to grasp the nylon loop attached to the tip of the spring and then pulled one in front and was fixed to the colon wall opposite the lesion (Figure 2C,D). This traction force allowed for adequate visualization of the submucosal cutting line, which resulted in a fast and safe dissection (Figure 2E). After the dissection, the S-O clip was detached from the colon wall and the specimen was collected (Figure 2F).

### 2.5. Evaluation of Therapeutic Efficacy and Complications

The surgery duration, dissection speed, complete resection rate, perforation rate, and bleeding rate were compared between the two groups and assessed separately for the right colon (i.e., transverse colon, ascending colon, and cecum) and the left colon (i.e., descending colon and sigmoid colon). The surgery duration was calculated from the initial mucosal incision to the end of submucosal dissection. The dissection time was defined as the time-lapse from the end of the circumferential mucosal cut to the end of submucosal dissection. The lesion area, which was approximated as an ellipse, was determined by measuring the major axis (A) and the minor axis (B). The resected area was calculated as πAB/4. The dissection speed was calculated by dividing the resected area by the duration of the dissection. Perforation was confirmed endoscopically during ESD, and free air was confirmed on abdominal computed tomography. Hemorrhage was defined as massive intraoperative bleeding that required blood transfusion or as postoperative bleeding that required hemostatic treatment such as endoscopic clipping or coagulation. 

### 2.6. Statistical Analysis

The chi-squared test was used for comparisons between categorical data, whereas the Mann–Whitney U test was used for comparing continuous data. A *p*-value of < 0.05 was considered statistically significant. StatView 5.0 (Abacus Concepts Inc., Berkeley, CA, USA) was used to perform all statistical analyses.

## 3. Results

From September 2015 to January 2020, 185 colorectal tumors underwent ESD at our hospital. We divided the patients into two groups according to the timing of ESD; Conventional ESD group (CO group, *n* = 66) or S-O clip-assisted group (SO group, *n* = 119). In CO group, 19 patients were excluded from enrollment because they had rectal lesion (*n* = 15), non-lifting sign (*n* = 2), and a lesion measuring over 50 mm (*n* = 2). In SO group, 39 patients were excluded from enrollment. Finally, analysis was performed on 47 and 80 patients in the CO and SO groups, respectively (Figure 3).

As shown in Table 1, the demographics and clinicopathologic features of the cases did not differ between the two groups. The overall outcomes are shown in Table 2. Compared with the CO group, the SO group had a significantly shorter surgery duration (73.9 ± 43.5 min vs. 52.3 ± 21.8 min; *p* = 0.0006), a significantly greater dissection speed (14.8 ± 8.7 min vs. 24.4 ± 12.9 min; *p* = 0.0014), and a significantly higher en bloc resection rate (80.9% vs. 98.8%; *p* ≤ 0.001). Overall, S-O clip-assisted ESD was able to reduce the procedure time of conventional ESD. No statistical significance in either group experienced massive hemorrhage or postoperative bleeding that required blood transfusion.

The results were analyzed separately for the right and left colon (Table 3). In the right colon, both surgery duration was significantly shorter and the dissection speed was significantly higher in the SO group than in the CO group; however, there was no significant difference in the lesion area between the two groups. Furthermore, the en bloc resection rate was significantly improved in the right colon. On the other hand, there was no such trend in the left colon as in the right colon.

## 4. Discussion

The maintenance of tension and good visibility of the submucosal layer is an important prerequisite for a fast and safe submucosal dissection. In surgery, the assistant usually maintains tension using a proper force to allow for easier tissue dissection. However, during ESD, it is not easy to maintain good traction because the endoscope has only one working channel for the electrical surgical knife. Therefore, a so-called “second hand” is necessary during ESD.

To achieve traction force during ESD, several methods have been developed. A distal hood with a transparent tip was the first device used to apply tension to the submucosal layer to enable the endoscope to easily enter the submucosal layer and to stabilize the electric knife during resection or dissection [14]. The use of a transparent hood with a small-caliber tip was reported to provide a clear field during submucosal dissection and for the control of bleeding [15]. A traction force can also be obtained simply by gravity without needing to use additional devices. The direction of the traction force can be controlled by changing the patient’s position [16]. However, when the flap is small at the first stage of submucosal dissection, gravity does not work sufficiently, and dissection becomes difficult for small lesions or those accompanied by fibrosis [17].

Several methods have been developed to generate a counter-traction to the lesion. The efficacy of the external forceps method [18,19] was reported for gastric and rectal ESD. In this method, traction is applied to the anal side using grasping vending forceps. This way, dissecting the submucosal layer of the grasped side can push or pull the lesion and make the submucosal layer more visible. However, it is difficult to send the forceps deep into the colon. Therefore, this method is limited for rectal ESD. The use of clips to achieve traction has been attempted by various methods. In the thread-and-clip method (Figure 4A), the clip is attached to the flap of the proximal lesion and the end of the thread is pulled to enable the lifting of the lesion during endoscope manipulation [20,21]. However, this method requires withdrawal and re-insertion of the colonoscope before applying the clip. In the clip-and-rubber-band method, a clip is used with a rubber band for continuous traction between the proximal side of the lesion and the normal mucosa [22,23]. The clip-and-ring-thread method (Figure 4B) can help pull the lesion to the intended direction to set a moderate traction force; the advantage of this method over the thread-and-clip one is its lower cost [24]. Since the thread itself has no contraction force, the traction force decreases as the lesion is dissected. Therefore, it is necessary to add a clip repeatedly to maintain the traction force.

The use of the S-O clip can allow for the pulling of the submucosal layer using the spring and can assist in the first phase of dissection during colorectal ESD. Unlike other traction methods, the S-O clip does not require extensive equipment or lengthy preparation and can be used anywhere, regardless of the location of the lesion, including the deep colon. It can be applied through the forceps opening without removing the endoscope, just like a normal clip. Moreover, traction can be applied continuously in the desired direction. The spring of the device has good and constant strength in both extension or contraction and does not cause excessive tension of the muscle layer and is not affected by peristalsis. Stable traction is applied throughout the procedure, and the lesion is automatically pulled. After resection, the lesion is clipped to the intestinal mucosa through a nylon (or silicon) loop, which prevents it from moving to other sites due to peristalsis. Therefore, after ESD, the lesion can be slowly retrieved after hemostasis of the blood vessels of the resected ulcer. In a prospective study, Ritsuno et al. reported that ESD using S-O clip was safe and rapid for en bloc resection of large superficial colorectal tumors [25]. Furthermore, the S-O clip was approved by the pharmaceutical affairs bureau as a medical device for ESD of all gastrointestinal tracts. The efficacy of the S-O clip in gastric ESD has also been reported [26].

Although previous studies on traction methods showed shorter duration of procedure and lower rate of complications, it is unknown which lesions are ideal for this method. Our study is the first to report that lesions located in the right-side colon are strongly associated with shorter treatment time. In general, it has been reported that colorectal ESD is more difficult in the right colon than in the left colon [27]. The reasons are: (1) the longer insertion length of the endoscope and poor operability, (2) the influence of respiratory fluctuation, and (3) the presence of flexure. In the conventional traction method, the traction force decreases as the dissection progresses. However, the major advantage of the S-O clip is that a constant traction force can be maintained throughout procedure, and it can be towed in the intended direction, which provides a stable visual field even in the right colon. In this study, CO group had a longer surgery duration in the right side than in the left side mainly due to difficulty in maintaining visual field and poor maneuverability (Table 3). However, SO group had a significantly faster dissection speed in the right-side colon, which was not observed in the left side colon. Based on these findings, S-O clip is more favorable for right-sided lesions, and application of this method for simple lesions in the left side colon may not be beneficial. Furthermore, we believe that this method can potentially shorten the procedure time and reduce the complication rate effectively when it is performed by less experienced endoscopists, but it may not be necessary for experts to routinely use the S-O clip in all cases. Given that there are cases in which S-O clip-assisted ESD is inapplicable or ineffective, endoscopists should develop their skills based on the conventional method, even in the presence of new technique.

There were several limitations to this study. First, it was a retrospective analysis carried out in a single-center setting, and the experience of the endoscopist and the difficulty of the cases were not uniform. Because it is a regional central hospital, most cases were difficult to treat; therefore, many large lesions of the right colon were treated. However, ESD using S-O clip showed a certain treatment effect to some extent, independent of the surgeon’s experience. Secondly, rectal lesions were excluded from this study. The reason for excluding rectal lesions was that the spring part of the S-O clip needed to be pulled to the anal side of the lesion, and the other side of the S-O clip may or may not be able to be fixed to the rectal wall. In general, rectal lesions have relatively large lesion sizes, long ESD procedure times, low perforation rates, and high bleeding rates. Therefore, the exclusion of rectal lesions may have affected the surgery duration and complication rate. Finally, the influence of the learning curve is discussed. Although we compared the learning curves for the entire study, we did not analyze the learning curves separately for experienced and less-experienced endoscopists because the number of cases for less-experienced endoscopists was small. In the future, we will carry out more studies with bigger samples to examine the usefulness of the less experienced and to develop a system for ESD training.

## 5. Conclusions

The S-O clip-assisted method for ESD shortened the surgery duration and increased the en bloc resection rate and dissection speed, especially in the right colon. Even endoscopists who had less experience in colorectal ESD were able to perform this procedure safely and rapidly. Therefore, S-O clip-assisted ESD can be the most suitable method for introducing colorectal ESD.

## Figures and Tables

**Figure 1 jcm-11-00141-f001:**
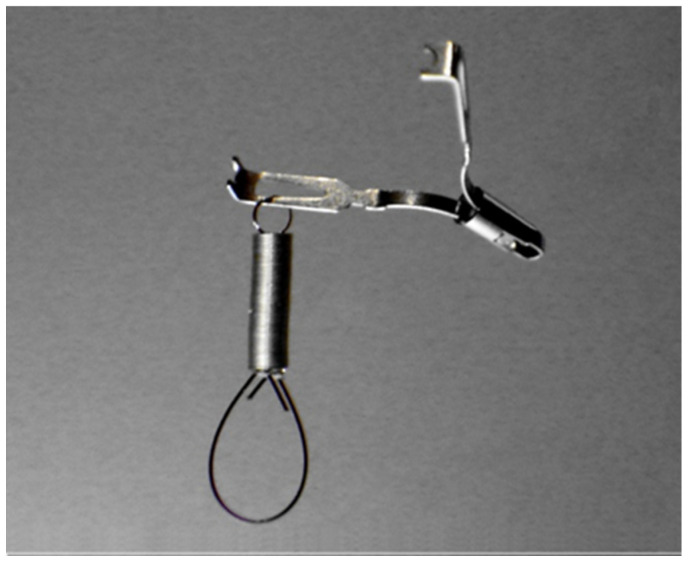
The external appearance of an S-O clip. The S-O clip comprised a metal clip (ZEOCLIP; Zeon Medical Co., Ltd.) and a 5 mm long spring. A nylon loop is attached to the other side of the spring and fixed to the colon wall using a second clip. The S-O clip can be passed through the channel of a conventional endoscope.

**Figure 2 jcm-11-00141-f002:**
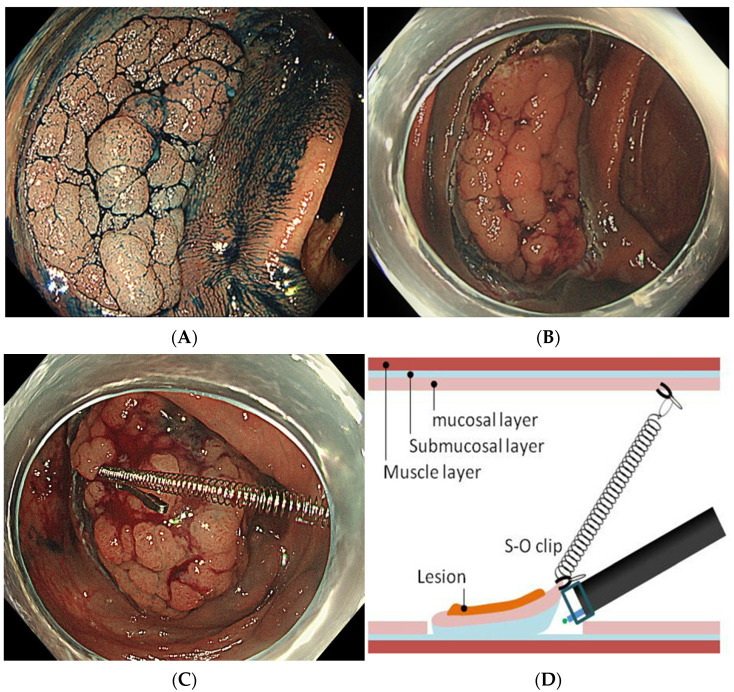
Method of S-O clip-assisted ESD. (**A**) An endoscopic examination with narrow-band imaging and 0.4% indigo carmine is conducted before ESD; (**B**) A circumferential incision of the mucosal layer is performed; (**C**,**D**) The S-O clip is attached to the proximal edge of the lesion, and another clip is used to grasp the nylon loop and pull one in front to fix to the colon wall opposite the lesion; (**E**) A counter-traction force allows good visualization of the submucosal cutting line; (**F**) Resected specimen.

**Figure 3 jcm-11-00141-f003:**
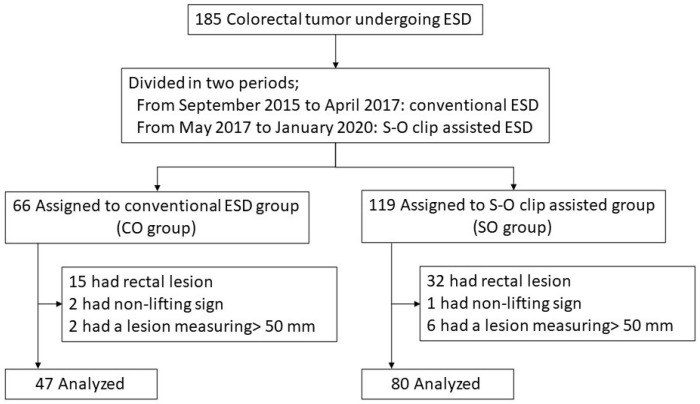
Flow diagram of the study patients.

**Figure 4 jcm-11-00141-f004:**
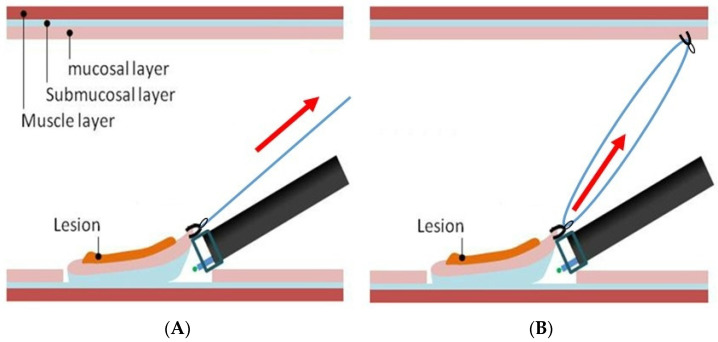
Method of traction-assisted ESD. (**A**) The thread-and-clip method. The clip is attached to the proximal lesion and the end of the thread is pulled to enable the lifting of the lesion; (**B**) The clip-and-ring-thread method. It can help pull the lesion to the intended direction to set a traction force.

**Table 1 jcm-11-00141-t001:** Patient demographics and clinicopathologic features.

	CO Group (*n* = 47)	SO Group (*n* = 80)	*p*-Value
Male/Female, *n*	32/15	47/33	0.345
Mean age (range), years	65.5 (38–80)	69.7 (39–89)	0.531
Lesion size, mean ± SD (range), mm	29.4 ± 9.1 (20–48)	30.6 ± 7.5 (20–50)	0.272
Lesion location			0.685
Right colon, *n*	35	56	
Left colon, *n*	12	24	

**Table 2 jcm-11-00141-t002:** Overall outcomes.

	CO Group (*n* = 47)	SO Group (*n* = 80)	*p*-Value
Surgery duration, mean ± SD (range), min	73.9 ± 43.5 (31–226)	52.3 ± 21.8 (16–113)	0.0006 *
Lesion area, mean ± SD (range), mm^2^	616.8 ± 576.8 (235.6–1507.9)	660.6 ± 333.6 (259.2–1696.4)	0.227
Dissection time, mean ± SD (range), min	49.7 ± 37.1 (17–189)	31.9 ± 16.4 (7–82)	<0.001 *
Dissection speed, mean ± SD (range), mm^2^/min	14.8 ± 8.7 (4.1–50.1)	24.4 ± 12.9 (5.5–70.6)	0.0014 *
En bloc resection rate, % (*n*)	80.9 (38/47)	98.8 (79/80)	<0.001 *
Perforation rate, % (*n*)	4.3 (2/47)	1.3 (1/80)	0.554
Hemorrhage rate, % (*n*)	0 (0/47)	2.5 (2/80)	0.530

* A *p* value of < 0.05 was considered statistically significant.

**Table 3 jcm-11-00141-t003:** Separate analysis for the left colon and the right colon.

	CO Group	SO Group	*p*-Value
Right colon, *n*	35	56	
Surgery duration, mean ± SD (range), min	78.1 ± 48.0 (33–226)	52.2 ± 21.3 (16–113)	0.0054 *
Lesion area, mean ± SD (range), mm^2^	648.4 ± 660.4 (235.6–1507.9)	685.5 ± 324.3 (259.1–1445.1)	0.1220
Dissection time, mean ± SD (range), min	51.5 ± 40.9 (17–189)	30.7 ± 15.2 (7–64)	0.0019 *
Dissection speed, mean ± SD (range), mm^2^/min	14.9 ± 9.1 (4.0–50.1)	25.4 ± 11.7 (9.5–61.7)	<0.001 *
En bloc resection rate, % (*n*)	77.1 (27/35)	98.2 (55/56)	0.0018 *
Left colon, *n*	12	24	
Surgery duration, mean ± SD (range), min	61.5 ± 24.1 (31–121)	51.9± 18.2 (26–112)	0.3139
Lesion area, mean ± SD (range), mm^2^	524.6 ± 175.4 (314.1–824.6)	563.0 ± 291.4 (311.0–1696.4)	0.9464
Dissection time, mean ± SD (range), min	44.4 ± 23.4 (20–100)	33.3 ± 18.1 (14–82)	0.1488
Dissection speed, mean ± SD (range), mm^2^/min	14.0 ± 8.0 (5.7–32.1)	22.0 ± 15.5 (5.4–70.6)	0.1587
En bloc resection rate, % (*n*)	91.7 (11/12)	100 (24/24)	0.3333

* A *p* value of < 0.05 was considered statistically significant.

## Data Availability

Data sharing is not applicable to this article.

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
