# Peer review of "Effectiveness of S-O Clip-Assisted Colorectal Endoscopic Submucosal Dissection"

_jcm, 2021, doi:10.3390/jcm11010141_

Round 1

Reviewer 1 Report

  • This is a single center retrospective analysis on the effectiveness of S-O clip assisted endoscopic submucosal dissection in patients with colon. Data of 127 patients (47 conventional and 80 following S-O clip assisted) were compared. S-O clip assisted ESD results in significant shorter surgery duration, faster dissection, higher en bloc resection rate and lower complication rate. 

    • Reported Patient numbers differ throughout the manuscript. Please correct.
    • Numbers are insufficient to conduct meaningful learning curve analysis. I would recommend removing this from the manuscript.

Author Response

Response to Reviewer 1 Comments (Manuscript ID: jcm-1492980)

Point 1: Reported Patient numbers differ throughout the manuscript. Please correct.

Response 1: Thank you for checking our manuscript thoroughly. The patient number has been corrected accordingly.

(Please see the attachment revised manuscript P.1, L.13-14)

We conducted a retrospective study on 185 patients who underwent colorectal ESD from January 2015 to January 2020.

Point 2: Numbers are insufficient to conduct meaningful learning curve analysis. I would recommend removing this from the manuscript.

Response 2: We hypothesized that there were two main factors that affected the main outcomes; introduction of S-O clip and improvement of ESD skills of endoscopists over the study period. Although we attempted to evaluate the effect of learning curve, it appears that the amount of ESD experiences had a little effect on the outcomes and the sample size was too small. We agree with the reviewer’s comment, therefore the paragraph has been removed as requested.

We are grateful for your constructive comments.

Reviewer 2 Report

In this paper, the Authors compare two submucosal endoscopic resection methods performed over a period of more than five years.

To improve the visibility of the surgical field and to ensure greater safety in carrying out endoscopic colorectal submucosal dissection, methods and tools were sought to reduce in particular the risk of perforations and bleeding, and to shorten the intervention time.

The Authors used from 2017 the S-O clip, a device that produces effective traction during the endoscopic maneuver improving the safety and timing of these interventions.

They retrospectively analyzed the data collected from 2015 to 2020, comparing the interventions performed with the conventional method and those with the use of the S-O clip, evaluating the averages of execution times and the frequencies of the main complications recorded.

The advantages found with the S-O Clip method are highlighted by using excessive statistical significance tests, taking into account that the description of the data collection is incomplete and confusing, in fact:

- different numbers are indicated in the abstract and in the Materials and Methods section;

- it is necessary that Authors specify the periods of use of the two methods,

- they must clarify the differences in the cases dealt with,

- they should explain the reasons for any further use of the conventional method in the presence of the innovative technique.

The Authors must indicate the ranges of variation observed in the measurements carried out with means and standard deviations.

The bibliography reported shows that this topic was already discussed a few years ago and therefore to support the benefits of the method with S-O clip it becomes important to update it.

Author Response

Response to Reviewer 2 comments (Manuscript ID: jcm-1492980)

Point 1: Different numbers are indicated in the abstract and in the Materials and Methods section.

Response 1: Thank you for checking our manuscript thoroughly. The number of patients has been corrected and the paragraph has been moved to the Results section to meet the request from reviewer 3 (please refer Comment 3).

(Please see the attachment revised manuscript P.1, L.13-14)

We conducted a retrospective study on 185 patients who underwent colorectal ESD from January 2015 to January 2020.

Point 2: It is necessary that Authors specify the periods of use of the two methods.

Response 2: The conventional method was performed from September 2015 to May 2017 and the SO method was performed from May 2017 to January 2020. This has been added to the Methods section to clarify the period of the two procedures.

(Revised manuscript P.2, L.46-50)

 In this retrospective study, patients underwent colorectal ESD at the Toyama University Hospital were divided into two groups before and after the introduction of S-O clips (Figure 2): the CO group underwent conventional ESD from September 2015 to April 2017, and the SO group underwent ESD with S-O clips from May 2017 to January 2020.

Point 3: They must clarify the differences in the cases dealt with.

Response 3: As the reviewer’s insightful comment, the selection or exclusion criteria for each group were unclear in the original manuscript. To solve this issue, we created a new figure containing a patient flow diagram (Figure 3) and inserted a detail description about the figure in the Results manuscript.

(Revised manuscript P.4, L.133-139)

 From September 2015 to January 2020, 158 colorectal tumors underwent ESD at our hospital. We divided the patients into two groups according to the timing of ESD; Conven-tional ESD group (CO group, n=66) or S-O clip assisted group (SO group, n=119). In CO group, 19 patients were excluded from enrollment because they had rectal lesion (n=15), non-lifting sign (n=2) and a lesion measuring over 50mm (n=2). In SO group, 39 patients were excluded from enrollment. Finally, analysis was performed on 47 and 80 patients in the CO and SO groups, respectively (Figure 3).

Point 4: They should explain the reasons for any further use of the conventional method in the presence of the innovative technique.

Response 4: Although we believe that S-O clip assisted ESD should be the standard approach to achieve safe and efficient ESD, rectal lesions may not be suitable for S-O clip as it may not be possible to receive traction force from the anal side, especially in the lower rectum cases. For such cases, ESD should be performed by conventional method. Therefore, we added the following sentence.

(Revised manuscript P.8, L.231-235)

 However, this method can potentially shorten the procedure time and reduce the complication rate especially when it is performed by less experienced endoscopists. On the other hand, we also believe that routine use of SO clip is unnecessary for simple cases performed by experts. Therefore, endoscopist should be able to perform conventional method in the presence of new technique.

Point 5: The Authors must indicate the ranges of variation observed in the measurements carried out with means and standard deviations.

Response 5: We presented most of measurement with means, standard deviations. Since range was missing in some of the measurements, we added ranges in the Revised manuscript P.5, Table 1, Table 2 and P.6, Table 3.

Point 6: The bibliography reported shows that this topic was already discussed a few years ago and therefore to support the benefits of the method with S-O clip it becomes important to update it.

Response 6: Although there are several reports that illustrates the efficacy of S-O clip, this is the first study that showed the S-O clip traction is more effective in the right-side colon. We modified the Discussion section to emphasize that S-O clip had a greater effect in the right-side colon.

(Revised manuscript P.7-8, L.224-231)

However, our results showed a shorten in treatment time, especially in the right colon. Previous reports on traction methods have mentioned shorter treatment time and safety, but this report is the first report to show shorter treatment time, especially in the right colon. The reason for this is that the S-O clip can maintain a constant traction force and continue to detach at a constant speed even when the lesion is large, which may have shortened the treatment time for the right colon. Furthermore, when the lesion is large, the detached mucosa itself acts as a flap, which may have shortened the treatment time, especially in the right colon where large lesions are common.

We are grateful for the constructive comments.

Reviewer 3 Report

I had the privilege to review this manuscript investigating the advantages of using the S-O clip during ESD for colorectal lesions. This is a retrospective study on 127 patients who underwent ESD for colorectal malignancies. Of these, 80 patients underwent ESD using the S-O clip and the remaining underwent standard ESD. Besides a few minor typos, the manuscript is well-written.

Please find below my comments:

INTRODUCTION

Line-31: please correct the form “colorectum” in colon and rectum.

Line-32: the statement “regardless of tumor size” requires citation. I have serious doubts that a bulky tumor could be excised via ESD. Morevoer, ESD is not indicated for any stage. In fact, patients with lesions measuring > 50mm have been excluded from the present study. I would advice the authors to reformulate the sentence to avoid giving an easily misunderstandable message. Eventually, current indications for colorectal ESD could be cited here.

Line-37: please change “easy” with easier.

MATERIALS AND METHODS

Why have the authors decided to exclude rectal lesions from this study? It would be useful to include the reason among the methods, as it can be an important source of bias.

Lines 44-57: the numbers pertaining to the colon tumors and patients operated should be moved to the results section. Moreover, I suggest removing the number 185, as it can be confounding to the reader, since only 127 patients have been considered for the study. Otherwise, the number of patients excluded for each exclusion parameter should be listed. A revision of this part of the methods is necessary.

DISCUSSION

Line 228: It is not clear to me what the authors claim as “certain treatment effect”. Can the authors please clarify this sentence?

An image including the various methods currently available for endoscopic counter-traction of the lesions undergoing ESD would be useful to the reader.

The authors have stated some of the positive features of the S-O clips, however, I believe that giving more emphasis on the practical differences between the S-O clips over the other counter-traction methods would greatly benefit the manuscript.

Author Response

Response to Reviewer 3 comments (Manuscript ID: jcm-1492980)

Point 1: Line-31: please correct the form “colorectum” in colon and rectum.

Response 1: We changed from colorectum to colon and rectum according to the comments.

(Revised manuscript P.1, L.30-31)

Endoscopic submucosal dissection (ESD) is an established treatment for intramucosal tumors of the gastrointestinal tract, including the colon and rectum.

Point 2: Line-32: the statement “regardless of tumor size” requires citation. I have serious doubts that a bulky tumor could be excised via ESD. Moreover, ESD is not indicated for any stage. In fact, patients with lesions measuring > 50mm have been excluded from the present study. I would advise the authors to reformulate the sentence to avoid giving an easily misunderstandable message. Eventually, current indications for colorectal ESD could be cited here.

Response 2: Thank you for your insightful comments. Although diameter of tumor is not a definitive parameter that determines the indication of colorectal ESD, we agree that there was a misleading in the original manuscript. In comparison to EMR, which allows en bloc resection for lesions measuring up to around 20mm in size, ESD enables en bloc resection of all size provided that the lesion has no deep submucosal invasion. In summary, this should be clarified by inserting "compared to endoscopic mucosal resection (EMR)" in the manuscript.

 The reason for excluding lesions measuring over 50mm is that the Japanese national insurance for colorectal ESD in 2012 was covering lesions measuring from 20 mm up to 50 mm.

(Revised manuscript P.1, L.31-33)

This method can enable en bloc resection, compared to endoscopic mucosal resection (EMR), and has a low recurrence rate of 0.4%–1.0%.

Point 3: Line-37: please change “easy” with easier.

Response 3: We changed word accordingly, thank you.

(Revised manuscript P.1, L.38-39)

 Performing traction-assisted ESD will be easier if the submucosal layer can be directly visualized after the mucosal cut.

Point 4: Why have the authors decided to exclude rectal lesions from this study? It would be useful to include the reason among the methods, as it can be an important source of bias.

Response 4: Coil spring of the SO clip requires traction from the anal side of the lesion, therefore it is physically impossible to fix the clip especially when the lesion is located in the lower rectum. Therefore, we uniformly excluded rectal lesions.

(Revised manuscript P.8, L.242-245)

 The reason for excluding rectal lesions was that the spring part of the SO clip needed to be pulled to the anal side of the lesion, and the other side of the SO clip may or may not be able to be fixed to the rectal wall.

Point 5: Lines 44-57: the numbers pertaining to the colon tumors and patients operated should be moved to the results section. Moreover, I suggest removing the number 185, as it can be confounding to the reader, since only 127 patients have been considered for the study. Otherwise, the number of patients excluded for each exclusion parameter should be listed. A revision of this part of the methods is necessary.

Response 5: According to comment, the number of patients has been moved to the Results section. We assigned 185 patients into two groups according to the period, and the number of excluded cases was noted. A patient flow was created and inserted as Figure 3 for clarification.

(Revised manuscript P.2, L.46-50)

 In this retrospective study, patients underwent colorectal ESD at the Toyama University Hospital were divided into two groups before and after the introduction of S-O clips (Figure 2): the CO group underwent conventional ESD from September 2015 to April 2017, and the SO group underwent ESD with S-O clips from May 2017 to January 2020.

(Revised manuscript P.4, L.133-139)

 From September 2015 to January 2020, 185 colorectal tumors underwent ESD at our hospital. We divided the patients into two groups according to the timing of ESD; Conven-tional ESD group (CO group, n=66) or S-O clip assisted group (SO group, n=119). In CO group, 19 patients were excluded from enrollment because they had rectal lesion (n=15), non-lifting sign (n=2) and a lesion measuring over 50mm (n=2). In SO group, 39 patients were excluded from enrollment. Finally, analysis was performed on 47 and 80 patients in the CO and SO groups, respectively (Figure 3).

Point 6: The authors have stated some of the positive features of the S-O clips, however, I believe that giving more emphasis on the practical differences between the S-O clips over the other counter-traction methods would greatly benefit the manuscript.

Response 6: Other counter-traction methods are described in the Disccusion paragraph. We created a new figure with illustrations to highlight the differences of each counter-traction method (Figure 4).

(Revised manuscript P.7, L.198-202)

Figure 4 was added.

Point7: Line 228: It is not clear to me what the authors claim as “certain treatment effect”. Can the authors please clarify this sentence? An image including the various methods currently available for endoscopic counter-traction of the lesions undergoing ESD would be useful to the reader.

Response 7: In the original manuscript, we included a learning curve analysis of S-O clip assisted ESD in the Discussion section. Given that the procedure time in the early period and the late period had no significant difference, the effectiveness of the S-O clip appears to be constant, regardless of experience of S-O clip usage. Based on this result, the authors commented that the use of S-O clip provides a "certain treatment effect". However, reviewer Comment 1 pointed out that the learning curve analysis is unreliable due to its small sample size, therefore we decided to remove the paragraph as requested.

We agree that the sentence "certain treatment effect" is inappropriate in the revised manuscript and should be removed. We deleted the sentence, and inserted Figure 4 instead, which illustrates the differences of other counter-traction methods.

We are grateful for the constructive comments.

Round 2

Reviewer 1 Report

This is the first revision of the authors' manuscript. The revised version sufficiently addressed my comments.

Author Response

Point: This is the first revision of the authors' manuscript. The revised version sufficiently addressed my comments.

Response: Thank you for your constructive suggestions. We have made sufficient corrections for all reviewer's comments.

Reviewer 2 Report

The Authors introduced changes in the original manuscript, following the suggestions of the referees, that make the work much more understandable. In the letter, they indicate that an innovative aspect of this analysis is the comparison between interventions carried out on the right colon compared to the left one. This topic must be further proposed in the final work.

Author Response

Response to Reviewer 2 comments (Round 2)

Point 1: The Authors introduced changes in the original manuscript, following the suggestions of the referees, that make the work much more understandable. In the letter, they indicate that an innovative aspect of this analysis is the comparison between interventions carried out on the right colon compared to the left one. This topic must be further proposed in the final work.

Response 1: Thank you for your constructive comment. To clarify the advantages of S-O clip as compared with other traction methods for colorectal ESD, the following paragraph was changed.

(Revised manuscript P.7-8, L.220-14)

 Although previous studies on traction methods showed shorter duration of procedure and lower rate of complications, it is unknown which lesions are ideal for this method. Our study is the first to report that lesions located in the right-side colon are strongly associated with shorter treatment time. In general, it has been reported that colorectal ESD is more difficult in the right colon than in the left colon [27]. The reasons are, (1) the longer insertion length of the endoscope and poor operability, (2) the influence of respiratory fluctuation, and (3) the presence of flexure. In the conventional traction method, the traction force decreases as the dissection progresses. However, the major advantage of the S-O clip is that a constant traction force can be maintained throughout procedure, and it can be towed in the intended direction, which provides a stable visual field even in the right colon. In this study, CO group had a longer surgery duration in the right side than in the left side mainly due to difficulty in maintaining visual field and poor maneuverability (Table 3). However, SO group had a significantly faster dissection speed in the right-side colon, which was not observed in the left side colon. Based on these findings, S-O clip is more favorable for right-sided lesions, and application of this method for simple lesions in the left side colon may not be beneficial. Furthermore, we believe that this method can potentially shorten the procedure time and reduce the complication rate effectively when it is performed by less experienced endoscopists, but it may not be necessary for experts to routinely use S-O clip in all cases. Given that there are cases which S-O clip assisted ESD is inapplicable or ineffective, endoscopists should develop their skills based on the conventional method, even in the presence of new technique.

We are grateful for the constructive comments.